# Ultrasound Morphometry and Mean Echogenicity of Digital Flexor Tendons, Suspensory Ligament, and Accessory Ligament of Digital Deep Flexor Tendon in Gaited Horses

**DOI:** 10.3390/ani13081411

**Published:** 2023-04-20

**Authors:** Jackson Schade, Anderson Fernando de Souza, Lorenzo Costa Vincensi, Thiago Rinaldi Müller, Joandes Henrique Fonteque

**Affiliations:** Department of Veterinary Medicine, Agroveterinary Sciences Center, Santa Catarina State University, Lages 2007, SC, Brazil

**Keywords:** equine, tendinopathies, diagnostic imaging, lameness

## Abstract

**Simple Summary:**

Breed variations related to the tendon and ligament dimensions of the palmar metacarpal region have been described in horses; however, they have not been demonstrated in gaited horses or between the fore- and hindlimbs. In this study, we evaluate and compare the sonographic characteristics of the digital flexor tendons and ligaments of the palmar/plantar of the metacarpal and metatarsal regions in gaited horses and also establish normal reference ultrasound values for Mangalarga Marchador and Campeiro horses. Thus, the values presented in this paper can be used as a reference for the ultrasound evaluation of the digital flexor tendons and ligaments in the limbs of horses of these breeds.

**Abstract:**

The objective of this study was to evaluate and compare the sonographic characteristics of the digital flexor tendons and ligaments of the palmar/plantar of the metacarpal and metatarsal regions in gaited horses, as well as to establish normal reference ultrasound values for Mangalarga Marchador (MM) and Campeiro breeds. Transverse sonographic images were obtained of 50 adult and healthy horses from the MM (*n* = 25) and Campeiro (*n* = 25) breeds. The images were taken in six metacarpal/metatarsal zones, and the following measurements were taken: transverse area, circumference, dorsopalmar/plantar length, lateromedial length, and mean echogenicity. Differences were observed between breeds in the fore- and hindlimbs, and, even when not significant, the Campeiro breed tended to have higher values for most variables and structures. Variations between zones and between structures in the same zone followed a similar trend in both breeds for all variables. In addition, the dimensions and variations between zones and structures were different between the fore- and hindlimbs, highlighting the need for specific values for the digital flexor tendons and ligaments of the metatarsal plantar region. In conclusion, the digital flexor tendons, suspensory ligaments, and accessory ligaments of the digital deep flexor tendon are influenced by the breed factor in gaited horses, and they are different between the fore- and hindlimbs.

## 1. Introduction

Gaited horses are characterized by their high performance and comfort [1]. In Brazil, the Mangalarga Marchador (MM) breed is the most widely used breed for leisure, farm work, and various types of equestrian sports [2]. The Brazilian Association of Mangalarga Marchador Horse Breeders (Associação Brasileira dos Criadores do Cavalo Mangalarga Marchador—ABCCMM) is the largest in Latin America, and it has horses distributed in several countries around the world [3]. The Campeiro breed is the only breed of gaited horse that originated in southern Brazil, and it is considered locally adapted; therefore, it constitutes valuable genetic heritage. Thus, in addition to the great potential for horse riding, Campeiro horses are an important alternative for animal breeding programs [4].

Ultrasonography is the main tool used for the diagnosis of tendinopathies and desmopathies in horses [5]. Breed variations related to the sonographic characteristics of the digital flexor tendons, suspensory ligaments, and accessory ligaments of the digital deep flexor tendon are described in horses [6,7,8,9]. For example, Arabian horses have smaller tendon dimensions than Thoroughbreds, Swedish Standardbred Trotters, Turkish native horses, Anglo-Arabians, or Dutch Warmbloods [6]. In this way, gaited horses may present differences in relation to these characteristics, being influenced especially by the peculiar type of gait. However, there are few studies related to tendon and ligament evaluation in gaited horses, and only one study described some morphometric characteristics of the forelimb tendons of Mangalarga Marchador horses [10].

Knowledge of ultrasound characteristics and normal reference values is fundamental for accurate diagnosis, allowing the differentiation between normal and abnormal patterns [7,11]. Thus, the objective of the study was to evaluate and compare the morphometric characteristics of the digital flexor tendons, suspensory ligament, and accessory ligament of the digital deep flexor tendon in gaited horses, as well as to establish normal reference ultrasound values for these structures in MM and Campeiro breeds.

## 2. Materials and Methods

### 2.1. Ethical Statement

The study was approved by the Animal Use Ethics Committee (CEUA) of Santa Catarina State University (UDESC) under protocol number 5868161216.

### 2.2. Study Design

A cross-sectional study of ultrasound morphometry of the digital flexor tendons and ligaments of the metacarpal/metatarsal palmar region in gaited horses.

### 2.3. Horses

Twenty-five MM horses and twenty-five Campeiro horses were evaluated and selected to form homogeneous groups regarding body size and weight within the breeds and between breeds. The group of horses of the MM breed was represented by sixteen females, six stallions, and three geldings, with average age of 7.1 ± 3.3 years (2.5 to 17.0 years), wither height of 1.47 ± 0.03 m (1.40 to 1.52 m), body weight of 384.00 ± 39.93 kg (327.00 to 474.00 kg), body mass index (BMI) of 178.81 ± 17.18 kg/m^2^ (155.22 to 217.50 kg/m^2^), metacarpal circumference of 17.59 ± 0.54 cm (17.00 to 19.00 cm), and metatarsal circumference of 19.43 ± 0.61 cm (18.50 to 20.50 cm). The Campeiro breed was represented by eighteen females, six stallions, and one gelding, with an average age of 7.2 ± 3.6 years (3.0 to 18.0 years), wither height of 1.45 ± 0.02 m (1.39 to 1.48 m), body weight of 394.60 ± 36.87 kg (321.00 to 459.00 kg), BMI of 188.64 ± 18.52 kg/m^2^ (156.69 at 222.45 kg/m^2^), metacarpal circumference of 18.10 ± 0.69 cm (17.00 to 19.50 cm), and metatarsal circumference of 19.58 ± 0.69 cm (18.50 to 21.00 cm). The animals came from four properties located in the cities of Tijucas do Sul (MM) in the state of Paraná and three properties in the city of Curitibanos (Campeiro) in the state of Santa Catarina (all in south Brazil) (Figure 1). The owners of the horses were informed about the nature of the research, and they received, for their appreciation and signature, a Free and Informed Consent Form. Thus, the animals were included in the project only with the consent of their owners.

### 2.4. Inclusion Criteria

Only horses enrolled in the Studbook of ABCCMM and the Brazilian Association of Campeiro Horse Breeders (Associação Brasileira de Criadores de Cavalo Campeiro–ABRACCC) that present the gaits in their different forms were included in the study. To minimize the effects of physical activity on tendons and ligaments, only horses that had not been in physical training for at least six months were used [7]. Only clinically healthy animals with no history of injuries and no clinical manifestations related to the digital flexor tendons and ligaments of the metacarpal/metatarsal palmar/plantar region or lameness of any kind were admitted for an ultrasound examination.

### 2.5. Clinical Examination

The horses underwent a specific clinical examination of the locomotor system [12], which included history and static and dynamic inspection while walking, gaiting in a straight line, and gaiting in circles. Limb palpation was performed with special focus on the palmar/plantar structures of the metacarpal/metatarsal, initially with the limb resting on the ground and then with the lifted limb. Only animals that did not present any local alterations or lameness (grade 0/5 on the AAEP scale) were used for the sonographic evaluation. All clinical evaluations were performed by a single individual (JS).

### 2.6. Preparation and Ultrasound Examination

To ensure better acoustic coupling during the ultrasound examination, trichotomy (blade 40) of the palmar/plantar region of the metacarpal/metatarsal was performed, as well as the lateral and medial sides of the distal third, followed by washing the region with water and soap. Alcohol (70%) was firmly applied to the skin with the aid of a compress, followed by the use of specific gel between the skin and the probe during the exam. Untamed horses and those that did not allow any preparation or examination to be performed were previously sedated by intravenous administration of 10 to 20 μg/kg detomidine hydrochloride (Detomidin^®^).

A real-time ultrasound examination was performed with the horses in the standing position using the Sonoscape A6 Vet^®^ portable ultrasound device equipped with a 5–12 MHz multifrequency linear probe. Transverse ultrasound images of the palmar/plantar region of the metacarpal/metatarsal were recorded in six distinct zones [7,13]. For this, the metacarpal and metatarsal lengths of 10 horses of each breed were measured, and the average value was divided by six, determining the extent of each zone in each breed. In the forelimbs (FL), the measurement was performed between the distal aspect of the accessory carpal bone and the proximal surface of the lateral proximal sesamoid bone (PSB). For HL, the measurement was performed between the head of the fourth metatarsal bone and the proximal surface of the ipsilateral PSB. Each zone assigned to the examination was marked with white- or blue-colored chalk according to the coat color on the side of the metacarpal/metatarsal to facilitate its location during the examination. The transverse images were recorded in the middle portion of each zone (Figure 2), and the structures evaluated consisted of the superficial digital flexor tendon (SDFT), deep digital flexor tendon (DDFT), accessory ligament of digital deep flexor tendon (ALDDFT), and suspensory ligament (SL) (Figure 3). In the HL, transverse images of the proximal zones of DDFT and ALDDFT (zones 1, 2, and 3), and SL (zones 1 and 2) were obtained with the probe positioned on the plantaromedial surface of the metatarsus. In the most distal areas, the branches of the SL cannot be seen on the palmar/plantar side and, therefore, were evaluated on the lateral and medial palmar/plantar sides at three heights: (1) at the level of the SL bifurcation; (2) at the midpoint between the SL bifurcation and ipsilateral PSB; (3) at the insertion region of each branch of SL in PSB.

All ultrasound images were performed at a frequency of 8–12 MHz and a depth of 49 mm. Time gain compensation (TGC), focus, and gain controls were standardized and constant across all evaluations. The exams were performed by an equine veterinarian with five years of experience in equine diagnostic imaging (JS), and the images were recorded and stored on an external hard disk for later measurement.

### 2.7. Variables and Measurements

The measured variables were transversal area (TA; mm^2^), circumference (C; mm), dorsopalmar/plantar length (DP; mm), lateromedial length (LM; mm) [8], and mean echogenicity (ME). The determination of ME was performed via image analysis according to a scale with 256 shades of gray, where 0 = black and 255 = white [14]. Each variable was measured three times in each structure and each zone, and the mean value obtained was used to calculate the general mean. All measurements were performed by a single individual (JS) using ImageJ^®^ software [14].

### 2.8. Data Analysis

Descriptive data analysis was performed by calculating the arithmetic means and standard deviations of the morphometric and ME variables of the tendons and ligaments. Three measurements were performed for each structure in each zone, for which the coefficient of variation (CV) was calculated, and the measurements were repeated when the CV was greater than 5%. The reference values were calculated as the arithmetic mean, standard deviation, and 95% confidence interval based on the values obtained for the right fore- and hindlimbs. The normality of the data was evaluated using the Shapiro–Wilk test. To compare the mean values of the morphometric and ME variables between the different zones of each structure and between the different structures in each zone, the two-way ANOVA mixed model was used. The random factor ‘horse’ and the fixed factors ‘zones’ and ‘structure’ were tested. In cases of rejection of the null hypothesis, the Bonferroni test was applied. The student’s t-test was used to compare morphometric variables and mean echogenicity between breeds, ages, contralateral limbs, and between FL and HL. Analyses were performed using Graphpad Prism 7 software, with a significance level of 5%.

## 3. Results

### 3.1. Zone Determination

For the MM breed, mean values of 22.8 cm and 24.6 cm in length were found for the palmar and plantar regions of the metacarpal and metatarsal, resulting in six zones of 3.8 cm and 4.1 cm, respectively. Campeiro horses presented mean values of 22.8 cm and 24.0 cm in length for the palmar and plantar regions of the metacarpal and metatarsal, resulting in six zones of 3.8 cm and 4.0 cm, respectively.

### 3.2. Groups

Groups were formed to compare the values of the variables studied in each structure and zone, considering the contralateral limbs, FL, and HL in both breeds. Regarding age, two groups were formed for each breed: MM Group (1) nineteen horses aged ≤ 7 years, and Group (2) six horses > 7 years old; Campeiro Group (1) thirteen horses aged ≤ 6 years, and Group (2) twelve horses aged > 6 years. The division of age groups was based on values above and below the median age in each breed.

### 3.3. Normal Ultrasound Values

The mean values and standard deviations of the morphometric and ME variables of the digital flexor tendons and ligaments of the palmar/plantar regions of the metacarpals/metatarsals of the MM and Campeiro breeds are shown in Table 1 and Table 2, respectively. Values are shown for each structure in each zone of the right metacarpal and metatarsal.

### 3.4. Differences between Zones

The values related to the significant mean difference (*p* < 0.05) between the zones of the digital flexor tendons and the ligaments of the metacarpal palmar region in horses of the MM and Campeiro breeds are shown in Table 3 and Table 4, respectively. Variations in the dimensions and ME of the structures between the different metacarpal zones showed a similar tendency in both breeds, so the description is given in general, and the differences are mentioned only when pertinent.

As a general trend, SDFT showed a significant reduction in the TA in the first zones, then gradually increased in the distal zones. The DP length demonstrated an inverse reduction to the C and LM length, which increased distally. Regarding the ME, the SDFT was more echogenic in the proximal zones.

For the values of the TA, C, and LM length, the SDFT presented a reduction in the first zones, then gradually increased in the distal zones. The values of the DP length were similar in the proximal zones, increasing significantly and reaching higher values in zone 5. In the MM breed, zone 2 was the only one that showed a difference in relation to the ME, as it was the most echogenic. For the Campeiro breed, the proximal zones (1, 2, and 3) were more echogenic.

The ALDDFT presented a reduction in the values of the TA and DP length in the distal direction. Regarding C, there was no variation in either breed, except between zones 2 and 3 in the Campeiro breed. For the LM length, zone 1 was larger and different from all other zones. There was no variation regarding the ME between zones in the Campeiro breed, and, for the MM breed, it was higher in zone 2, which differed only from zones 1 and 4.

When assessing the SL, higher values of the TA alone were observed for zone 1. There were no variations between zones regarding the C, DP, and LM length for the MM breed, except for the LM length between zones 1 and 3. In the Campeiro breed, only zone 1 was larger in relation to the C and LM length. Little variation was observed for the ME in the Campeiro breed, and only zone 3 was more echogenic compared to zone 1. For the MM breed, zones 2 and 3 were more echogenic. The LB-SL and MB-SL showed a significant increase in all variables distally.

The values related to the significant mean difference (*p* < 0.05) between the zones of the digital flexor tendons and the ligaments of the metatarsal plantar region in horses of the MM and Campeiro breeds are shown in Table 5 and Table 6, respectively. Variations in the dimensions and ME of the structures between the different metacarpal zones showed a similar tendency for both breeds, so the description is given in general and the differences are mentioned only when pertinent.

In the HL, SDFT did not show any variation in TA values from zone 1 to zone 5, with higher values being observed only for zone 6. Regarding the DP length, there was an inverse reduction to the C and LM length that increased distally. There was no variation between zones in relation to the ME for the Campeiro breed, and, in the MM breed, less echogenicity was observed for zone 4, which differed only from zones 2 and 6.

Regarding the DDFT, there was a minor variation in the TA and C in the proximal zones, which gradually increased in the distal zones. For the Campeiro breed, the C presented higher values in zone 1 compared to zones 3 and 4. The DP length was shorter in zone 1, which increased and stabilized in zones 2, 3, and 4, and again increased and reached higher values in zone 5 for the MM breed. Similar values were observed for the DP length in the proximal zones for the Campeiro breed. As for the LM length, there was a tendency to decrease in the first zones and then increase in zones 5 and 6. The ME was higher in zone 1 and in the distal zones.

ALDDFT showed no variation between zones regarding the TA and DP length in the MM breed. In the Campeiro breed, the DP length differed only between zones 1 and 4, whereas in relation to the TA, zones 3 and 4 differed from zone 1, and zone 2 was different from zone 4. As for the C, zones 3 and 4 differed from zone 1, and zone 2 was different from zone 4 in the MM breed; for the Campeiro breed, only zones 1 and 2 were different. The LM length varied between zones, tending to decrease distally. However, zones 3 and 4 did not differ in either breed, nor did zones 1 and 2 for the MM breed and zones 2 and 3 for the Campeiro breed. The ME was homogeneous between the zones for both breeds, except for zone 3 in the MM breed, which was higher compared to zone 2.

When evaluating the SL, it can be observed that the TA, C, and LM length were higher only for zone 1. Regarding the DP length, zone 1 was larger than the other zones, and zone 4 was larger than zones 2 and 3 in the MM breed and larger than zone 2 in the Campeiro breed. The ME values were homogeneous among all zones. The LB-SL and MB-SL demonstrated an increase in the values of all variables between distal zones.

### 3.5. Differences between Structures

The values related to the significant mean difference (*p* < 0.05) of the morphometric and ME variables between the different structures in each zone of the metacarpal palmar region for the MM and Campeiro breeds are shown in Table 7 and Table 8, respectively.

Regarding the TA values in the FL, the SL was higher in zones 1, 2, 3, and 4, followed by the DDFT in both breeds. However, the DDFT did not differ from the SL in zone 1 for the MM breed. The SDFT and ALDDFT were similar in zones 1 and 2, and in zones 3 and 4, the ALDDFT had lower values than all other structures in the MM breed. For the Campeiro breed, the structure with the lowest TA in zones 1 and 2 was the SDFT, and in zones 3 and 4, the ALDDFT. In the most distal zones (5 and 6), the DDFT was higher than the SDFT. No difference was observed between the LB-SL and the LB-SL in any of the zones.

For the C values in the FL, the SL was higher in zones 1, 2, and 3 in both breeds but did not differ from the DDFT in zone 1 or the ALDDFT in zones 2 and 3 in the MM breed. For the Campeiro breed, the SL presented a C like the ALDDFT in zone 2 and like the ALDDFT and SDFT in zone 3. The SDFT showed lower values in zone 1, progressing to the highest value in zones 4, 5, and 6, reversing its values in relation to the DDFT, which was the smallest structure in zones 3, 4, 5, and 6 in both breeds. The MB-SL was higher compared to the LB-SL in zone 3 for the MM breed and zone 1 for the Campeiro breed.

When evaluating the values for DP length in the FL, the DDFT was the largest structure, followed by the SL and SDFT in zones 1, 2, 3, and 4, with similar values for the DDFT and SL in zone 4. The ALDDFT was the smallest structure from zone 1 to zone 4, and, in the most distal zones (5 and 6), the DDFT was larger than the SDFT. The LB-SL was higher than the MB-SL in zones 1 and 2. All differences related to DP length were similarly observed in both breeds.

Regarding the LM length in the FL, higher values were observed in zone 1 for the ALDDFT, SL, and DDFT, which were similar in both breeds, except for the DDFT in the Campeiro breed. In zone 2, the SL was higher, followed by the SDFT in the MM breed. In the Campeiro breed, on the other hand, the SL and ALDDFT showed higher values in zone 2, which did not differ from each other. The SDFT was the smallest structure in zone 1, progressing to have the largest structure in zone 3. This observation occurred inversely to the DDFT, which was the smallest structure in zone 2 in both breeds. No differences were observed between the LB-SL and MB-SL.

Regarding the ME, the DDFT and SDFT were the most echogenic structures in zone 1 in both breeds; however, for the Campeiro breed, the ALDDFT also presented similar values. In this zone, the SL was the structure with less echogenicity. From zone 2 to zone 4, the DDFT and ALDDFT were more echogenic, and in zone 3, the SL was similar. From zone 2, the SDFT was the least echogenic structure, and in zones 5 and 6, it showed less echogenicity in relation to the DDFT. The LB-SL was more echogenic compared to the MB-SL in all zones for the MM breed and only in zone 1 for the Campeiro breed.

The values related to the significant mean difference (*p* < 0.05) of the morphometric and ME variables between the different structures in each zone of the metatarsal plantar region for the MM and Campeiro breeds are shown in Table 9 and Table 10, respectively.

Through the analysis of the TA values in the HL, it can be evidenced that the SL is the largest structure in zones 1, 2, 3, and 4, followed by the DDFT and SDFT in the MM breed. In the Campeiro breed, a similar trend was observed. However, the DDFT and SL were like zone 2. The ALDDFT was the smallest structure in all zones (zones 1 to 4), and the DDFT was larger than the SDFT in zones 5 and 6. No differences were observed between the LB-SL and MB-SL in either breed.

Regarding the C in the HL, the LS presented higher values in zones 1 and 2, followed by the DDFT and SDFT in both breeds, except that the DDFT was similar to the SL in zone 2 for the Campeiro breed. In zones 3 and 4, the SDFT started to acquire higher values, followed by the SL and DDFT in the Campeiro breed, with a similar presentation observed for the MM breed, except for zone 3, where the SL was similar to the SDFT. Similarly, to the TA, the ALDDFT was the smallest structure in all zones. However, the SDFT was larger than the DDFT in zones 5 and 6 in both breeds. The LB-SL was higher than the MB-SL in zones 1 and 3 only for the Campeiro breed.

Regarding DP length, the SL showed higher values in zone 1, followed by the DDFT and SDFT in both breeds. From zone 2, the DDFT showed higher values, followed by the SL and SDFT in both breeds, except in zone 2 for the MM breed, in which the DDFT and SL were similar. In zones 5 and 6, the DDFT presented higher values than the SDFT, and the ALDDFT was the structure with the shorter DP lengths in all zones. The LB-SL showed higher values than the MB-SL in zone 2 in the MM and Campeiro breeds.

For the LM length, the SL presented higher values in zone 1, followed by the DDFT and SDFT in both breeds. From zone 2, the SDFT began to show longer LM lengths in the Campeiro breed, followed by the SL and DDFT, which were similar. In the MM breed, the SDFT, SL, and DDFT did not differ from each other in zone 2, and from zone 3, the SDFT became the largest structure, followed by the SL and DDFT. The ALDDFT was again the structure with lower values from zone 1 to zone 4, and the SDFT showed higher values when compared to the DDFT in zones 5 and 6 in both breeds. For the MM breed, the LB-SL presented higher values than the MB-SL.

A comparison of the ME values in the PLs showed that the ALDDFT was the most echogenic structure in all zones (1 to 4). However, in zone 1, it was similar to the DDFT, and in zone 2 it presented values similar to the DDFT and SL in the MM breed. The second most echogenic structure was the DDFT, which was equal to the SL in zones 1, 3, and 4 for the MM breed. For the Campeiro breed, like the MM breed, the DDFT was the second most echogenic structure. However, it was similar to SL in zones 3 and 4. The SDFT was the least echogenic structure in both breeds, and in zones 5 and 6, it also showed lower values compared to the DDFT. The LB-SL was more echogenic in zones 1 and 3 for the MM breed and only in zone 3 for the Campeiro breed.

### 3.6. Differences between Groups

No significant differences (*p* < 0.05) were observed for morphometric and ME variables between age groups and between right and left contralateral limbs, for both the FL and HL in both breeds.

The results regarding the comparison between the FL and PL in horses of MM and Campeiro breeds are shown in Table 1 and Table 2, respectively. Regarding the SDFT, a greater DP length was observed in zone 1 of the FL and a greater length of the LM in the four proximal zones of the PLs in both breeds. The DDFT presented higher TA and C values in zones 2, 3, and 4 of the HL, as well as greater DP and LM length values in the first four zones of the HL, except for the LM length in zone 1 for the MM breed. Regarding the ALDDFT, higher values were observed for the FL in all variables and zones. The SL presented a higher TA in zone 1 of the HL for the Campeiro breed and in all zones for the MM breed. For the C, higher values were observed for zone 1 of the HL in both breeds, and in relation to DP length, differences were found in all zones of the SL, except for zone 4 in the Campeiro breed. The LM length was higher in zone 1 of the SL for the MM breed and in zones 1 and 2 in the Campeiro breed. It is noteworthy that the differences observed in zone 1 of the SL were observed in all variables, with values that were higher and of a greater magnitude for the HL. Higher values of the TA and DP length were observed in zone 3 of the MB-SL and LB-SL of the HL in both breeds, and higher values of LM length in zones 2 and 3 of the PL LB-SL in the Campeiro breed.

The comparison related to the ME shows that the SDFT was more echogenic in the FL in the first zones in both breeds, and the DDFT presented a higher value for the HL in zones 1, 2, 3, 4, and 5 in the MM breed and in zones 1, 2, 3, and 4 for the Campeiro breed. The ALDDFT and SL showed higher echogenicity in the FL in all zones in both breeds. Only the differences that were consistent and with tendency were described. Thus, the differences between the FL and HL that occurred in a specific way can be observed in Table 1 and Table 2.

Through the comparison between breeds, significant differences (*p* < 0.05) can be observed, with higher values for the Campeiro breed, except for the C of the LB-SL, the DP length of the SL in the PL, and the DDFT and ALDDFT in the FL, which were larger in some zones in the MM breed (Table 11). The SDFT showed differences regarding the TA and LM length in all HL zones and most FL zones. The DDFT differed with respect to the DP length in the FL and the LM length in the HL, except for in zone 5. For the ALDDFT, the LM length was different in FL zones 1, 2, and 3, and in HL zones 1 and 3, which also differed from the C. The SL showed a difference for TA in all HL zones and in the DP length for all zones of the FL. The DP length in the HL was greater for the Campeiro breed only in zone 1, and in zones 2 and 3 it was greater for the MM breed. Regarding the ME, higher values were observed in the MM breed for distal areas of the DDFT in the FL and proximal zones of the ALDDFT in the FL and HL. Significant punctual differences that were not presented as a trend were not described and can be seen in Table 8. However, even without showing significant differences between the variables and structures, in general, a tendency towards higher values in the Campeiro breed can be observed (Table 2) when compared to the MM breed (Table 1).

## 4. Discussion

This is the first study to evaluate and compare the morphometric and ME characteristics of the digital flexor tendons, suspensory ligaments, and accessory ligaments of the digital deep flexor tendons of the fore- and hindlimbs between two breeds of gaited horses. All ultrasound evaluations and measurements were performed by a single individual to avoid variation in results between evaluators [15]. In addition, to determine intra-rater variation, seeking greater accuracy, each variable in each structure and zone was measured three times, for which a maximum CV of 5% was accepted [16]. To ensure normal values, all horses underwent a thorough clinical examination so that only non-lame horses, established through subjective evaluation, that were healthy in relation to the tendons and ligaments of the metacarpal/metatarsal palmar/plantar regions were used. In addition, to avoid the effects of training, only horses that had not been trained for at least six months were used [7]. The cellular and molecular components of flexor tendons and ligaments are known to undergo progressive changes with athletic training, which modify their biomechanical properties. These changes result in hypertrophy and a consequent increase in the TA that may remain for the life of the horse [17]. However, these effects were minimized as microlesions resulting from athletic training that could affect the determination of the TA and ME were avoided [18,19]. Thus, the morphometric values presented in this study can be used as normal reference values for ultrasound examination in horses of the MM (Appendix A) and Campeiro (Appendix A) breeds. Due to the great influence of factors, such as skin preparation [13], transducer gain level, inclination, and displacement [20], on ME values, these were used only for comparisons and, therefore, are not available as reference values.

For the evaluation and recording of the DDFT, ALDDFT, and SL images in the proximal areas of the HL, the probe displacement for the plantaromedial aspect of the metatarsus was necessary. In the other zones, all structures were visible in the plantar aspect, except the branches of the SL, as established by other authors for the FL [8]. An ultrasound evaluation of the proximal metatarsal region requires changes in the technique compared with the evaluation of the proximal metacarpal region due to anatomical differences in tendons and ligaments as well as the fourth metatarsal bone [5]. According to Rantanen et al. [13], these structures are adequately evaluated by the plantar aspect of zone 2 (2B in the authors’ description), which was not confirmed in the present study, possibly due to anatomical particulars of the evaluated animals. Whitcomb [5] considered the chestnut as an excellent anatomical marker for the location of the SL origin in the proximal metatarsal region. However, the chestnut was absent bilaterally in Campeiro horses and is of variable size and height between horses in both breeds, making its use as an anatomical marker unreliable. Therefore, it is suggested that the use of an invariable location structure, such as the head of the fourth metatarsal bone, could ensure greater accuracy for the localization of the proximal metatarsal tendons and ligaments.

Variations in the size and shape of the structures between the different metacarpal zones were generally shown to be similar in MM (Table 3) and Campeiro (Table 4) horses. These variations also tended to be similar to those found in the Icelandic [9] and Haflinger [8] horse breeds. However, the SL and ALDDFT showed no variation in relation to the C in the MM breed, and the SL did not vary in relation to the DP length in either breed. These findings did not follow the same pattern in the above breeds [8,9], except for the C of the SL in the Icelandic breed [9].

In the HL, variations related to the size and shape of structures between zones also showed a similar trend in both breeds studied (Table 5 and Table 6). However, they did not follow the same pattern observed in the FL, especially with respect to the proximal portion of the SDFT and DDFT and the ALDDFT and SL (Table 3 and Table 4). A comparison of these variations with other breeds was not possible with respect to the PL, due to the absence of studies that evaluated through ultrasound the flexor tendons and ligaments of the plantar region of the metatarsus. The variation in the dimensions of the structures between the different metacarpal zones demonstrates that the evaluation and comparison should be conducted in the same zone, with specific reference values for each one [6]. Thus, the present work demonstrates that this statement is also true for the evaluation of the flexor tendons and ligaments in the plantar region of the metatarsus.

The results observed by comparing the variables between the different structures in each zone of the palmar/plantar regions of the metacarpals and metatarsals showed a similar trend among horses of the MM (Table 7 and Table 8) and Campeiro (Table 8 and Table 9) breeds. However, the same trend was not observed between the FL and HL in each breed. The SL showed the highest TA when compared to other structures in the same zone, corroborating the results found in Purebred Spanish horses [7], Haflinger [8], and Icelandic horses [9]. However, with respect to other structures, divergent results were observed for all variables between breeds [7,8,9]. Thus, it can be suggested that the relationship between the dimensions of the digital flexor tendons and metacarpal ligaments may present variations influenced by the breed factor. However, the division of the zones was not performed in a standard way between the works, which may be responsible for the differences between breeds.

Differences related to the dimensions of digital flexor tendons and metacarpal ligaments between breeds have been described in several studies [6,7,8,9,21]. The present study demonstrates that these differences also occurred in the digital flexor tendons and palmar/plantar ligaments of the metacarpal and metatarsal regions between two breeds of gaited horses (Table 11), with a tendency to higher values for most variables in the Campeiro breed (Table 2) when compared to the MM breed (Table 1), both in the FL and HL. Until then, differences regarding these structures had not been evaluated or demonstrated in other studies for PLs between breeds.

The TA values observed in the digital flexor tendons of Thoroughbred horses, draft horses, and ponies were higher when compared to those found in the gaited horses evaluated in the present study [16]. In a study with Thoroughbred and Arabian horses, higher values were observed for Thoroughbreds in relation to most variables and structures when compared to values found in MM and Campeiro horses. The Arabian horses, in general, presented lower values, which were closer to those found for the MM breed [6]. Purebred Spanish horses presented higher TA values in relation to the SDFT, the DDFT, the proximal portion of the ALDDFT, and the SL. However, the values of the distal zones of the ALDDFT and the branches of the SL were smaller [7]. For Haflinger horses, similar values were observed for most variables and structures; however, the first zone of the SL presented higher TA values [8]. In a study conducted in Brazil with Brazilian Sport horses, Thoroughbreds, and Crioulo horses, the TA of the digital flexor tendons and the ALDDFT were measured and were higher in all breeds when compared to MM and Campeiro horses [21]. Lower values were observed for all morphometric variables in relation to the SDFT, DDFT, and ALDDFT in Icelandic horses. The SL presented C and TA values similar to those found in the MM breed, and the DP length was greater in the distal zones when compared with both breeds evaluated in the present study. Regarding the branches of the SL, lower values were observed in Icelandic horses, except for the LM length, which was higher in the first zone [9]. A single study related to the evaluation of the TA of the digital flexor tendons and palmar metacarpal ligaments in MM horses has been found in the literature [10]. However, unexpectedly, the values shown were higher than those found in the present study, except for zone 1 of the SL, which was smaller and increased abruptly in zone 2. In addition, images of the ALDDFT and SL were obtained up to zone 5. This divergence could be attributed to the length of the zones (3.5 cm) and the number of horses used (*n* = 15), which were smaller than those used in the present study. In addition, care related to inter- and intra-rater variation was not mentioned, which may be suggested by the standard deviation, which was, on average, 195% higher [10].

Variations between breeds regarding tendon and ligament dimensions have been attributed to the different physical constitutions between breed groups, such as height, body weight, and metacarpal diameter [7,16]. However, the horses evaluated in the present study had similar height, body weight, BMI, and metacarpal and metatarsal circumferences, and, regardless, differences were observed. Boehart et al. [9] attributed the differences to the stress generated by the special pattern of gait present in the Icelandic breed, emphasizing the importance of an ultrasound morphometric pattern in horses with a peculiar gait. Gaited horses, such as those of the MM and Campeiro breeds, have specialized movements characterized by repetitive and high-impact movements [3]. In addition, gait can be presented in different ways that have differentiated biomechanical characteristics [22]. Thus, it can be suggested that the differences found between the breeds studied and in relation to other breeds could be attributed, in part, to the peculiar characteristics of gait. In the present study, it was not possible to compare tendon and ligament morphometry between the different gait types present in the MM and Campeiro breeds. Therefore, future work related to the theme should be developed, aiming to compare these characteristics between gaited and non-gaited horses and between different gait types.

Studies evaluating the morphometric and sonographic characteristics of the digital flexor tendons and ligaments of the metatarsal plantar region in healthy horses are scarce [23]. In a study by Muylle et al. [24], the morphometric characteristics of the ALDDFT were determined by dissecting 165 specimens of the HL from slaughterhouses. The ligament was absent in 10/165 horses and, when present, was characterized as a single straight structure (143/155) or divided in part or in full into two to three bundles (12/155). Dimensions were variable, with a mean proximal width of 1.9 ± 1.2 cm (0.2–3.2 cm), a distal width of 1.2 ± 1.0 cm (0.2–2.6 cm), a proximal thickness of 1.3 ± 0.6 mm (0.1–3.3 mm), and a length of 14.0 ± 2.1 cm (7.9–18.5 cm). However, because they are slaughterhouse specimens, information that could influence the results, such as age, breed, horse size, and history of injuries, was not available. Dyson [23] evaluated the ultrasound characteristics of the ALDDFT in ten horses: six Warmbloods and four Thoroughbreds. In one of the horses, the ligament was bifid bilaterally, and there was variation in ligament thickness and shape among all horses studied. In the present study, the ligament was not visible in only one horse of the MM breed and was bifid only in the first zone in one horse of the MM breed and two of the Campeiro breed. In addition, it was only visible in zone 3 in a Campeiro horse. Ligament dimensions varied widely among horses, especially in proximal areas. Further studies are needed to better characterize the ALDDFT of the HL as well as between different breeds.

Regarding ME, few isolated differences between breeds were observed, which, therefore, are not considered different regarding this variable (Table 11). ME-related differences in structures between the different metacarpal zones had, except for some differences, a similar trend among the breeds studied (Table 3 and Table 4) and were similar to the results observed in the Spanish Purebred horses [7]. In HL, a similar trend was observed between breeds (Table 5 and Table 6), but it was different from that observed in the FL, especially in relation to the SDFT, which showed no variation in echogenicity between zones for the Campeiro breed and little variation for the MM breed. In the FL, the SDFT was more echogenic in the proximal zones and reduced its distal echogenicity, as observed by other authors [7,25]. The SL branches showed increased distal echogenicity in the FL and HL in both breeds.

When comparing the ME between the different structures in the FL (Table 7 and Table 8), it can be observed in both breeds that the DDFT and ALDDFT were the most echogenic structures from zone 2, disagreeing with the results found by Agut et al. [7], where the SL was the most echogenic. In a study conducted with newborn foals, Spinella et al. [14] also observed that the DDFT and ALDDFT were more echogenic than the SL and attributed the finding to the higher muscle fiber content in the constitution of the ligament in this age group. However, according to Reef [26], the DDFT and ALDDFT are commonly the most echogenic structures of the palmar metacarpal region. In these zones and in the distal zones (5 and 6), the least echogenic structure was the SDFT. However, in the first zone, the SL presented lower echogenicity, similar to that observed in Spanish Purebred horses and neonate foals [7,14]. Possibly, the variable amount of muscle fibers in the constitution of the SL contributed to its lower echogenicity in the proximal region [27]. Regarding the SL branches, a higher echogenicity was observed for the LB-SL in both breeds, unlike that observed in Spanish Purebred horses, in which the MB-SL was the most echogenic [7]. In the HL (Table 9 and Table 10), the most echogenic structure was the ALDDFT, followed by the DDFT and SL, and, similar to that observed in the FL, the SDFT was the structure with the lowest echogenicity in all zones.

Age-related differences were not observed with respect to morphometric and ME variables in tendons and ligaments of the palmar/plantar regions of the metacarpals and metatarsals, similar to the results found in the FL in other studies [9,14,28]. However, this influence has been observed in studies conducted with other breeds [7,8]. In agreement with previous information, there were no differences related to morphometric variables and ME between contralateral limbs [7,9,14,28]. Therefore, all comparisons and reference values demonstrated were based on the results obtained for the right FL.

As limitations of this study, we can highlight that the ultrasound study was performed by only one evaluator, and it was not possible to verify whether the observations made by other observers would be similar. In addition, only non-training horses were used, so the characteristics seen in the image may have differences compared to horses in training.

The present study demonstrates differences related to the morphometric and ME characteristics of tendons and ligaments between two horse breeds and between the FL and HL. The importance of evaluating tendons and ligaments in the plantar metatarsal region has been demonstrated since lesions also occur in this region [23,29,30,31,32]. In addition, the distribution of injuries may also be affected by developed athletic activity, increasing, in some cases, the frequency of injuries in the digital flexor tendons and ligaments in the HL [33]. Thus, studies related to ultrasound evaluations of these structures in HL are required, as are specific reference values for different breeds.

## 5. Conclusions

It is concluded that the Campeiro breed has digital flexor tendons, suspensory ligaments, and accessory ligaments of digital deep flexor tendons with larger dimensions than the Mangalarga Marchador breed. Variations related to size and echogenicity between zones and structures are similar between breeds in the fore- and hindlimbs. The size and echogenicity of the tendons and ligaments of the fore- and hindlimbs are different, as are the variations between the zones and structures. This study demonstrates that specific values should be used for ultrasound evaluations of the digital flexor tendons and ligaments in the fore- and hindlimbs of gaited horses. Therefore, the available data can be used as an aid for the ultrasound diagnosis of tendinopathies and desmopathies in horses of the Mangalarga Marchador and Campeiro breeds.

## Figures and Tables

**Figure 1 animals-13-01411-f001:**
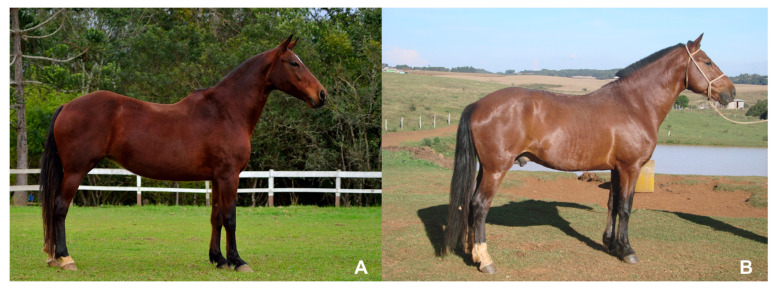
Images showing specimens of the Mangalarga Marchador (**A**) and Campeiro (**B**) horses included in the study.

**Figure 2 animals-13-01411-f002:**
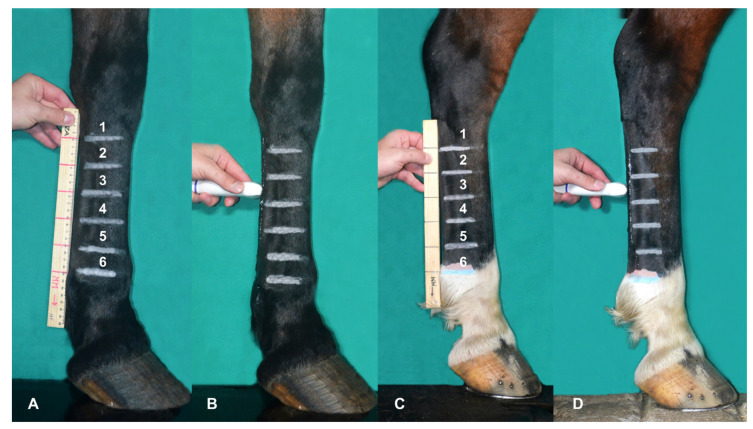
Determining the zones and positioning the probe for forelimbs (**A**,**B**) and hindlimbs (**C**,**D**).

**Figure 3 animals-13-01411-f003:**
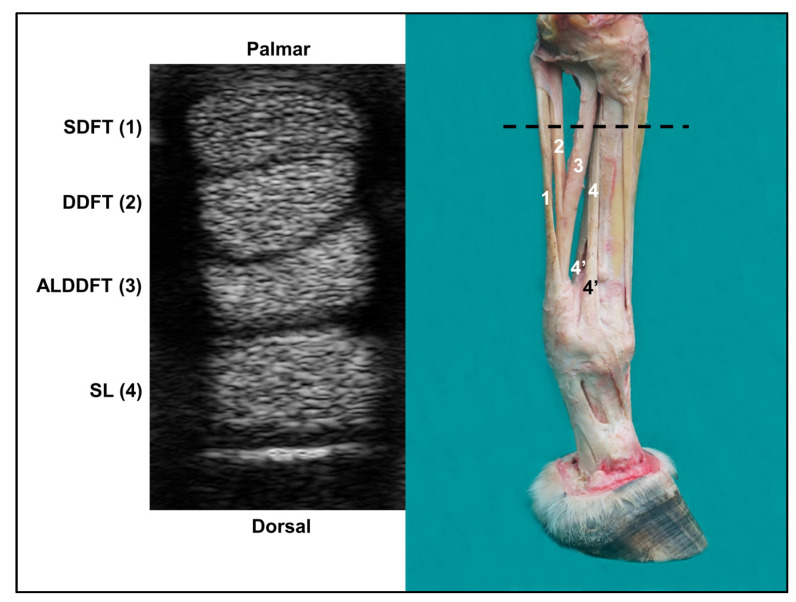
Dissected specimen of the forelimb and ultrasound image (zone 2, at the level of the dashed line) demonstrating the evaluated structures. SDFT: superficial digital flexor tendon (1); DDFT: deep digital flexor tendon (2); ALDDFT: accessory ligament of digital deep flexor tendon (3); SL: suspensory ligament (4); 4’: SL branches.

**Table 1 animals-13-01411-t001:** Means and standard deviations of the morphometric variables and mean echogenicity of the digital flexor tendons and ligaments of the metacarpal palmar/plantar regions in 25 Mangalarga horses.

Structure	Zone	AT (mm^2^)		Circumference (mm)		DP Length (mm)		LM Length (mm)		Mean Echogenicity	
		FL	HL	FL	HL	FL	HL	FL	HL	FL	HL
SDFT	1	65.6 ± 6.3	62.5 ± 4.9	32.1 ± 1.4	32.2 ± 1.6	6.4 ± 0.6	5.7 ± 0.4 *	11.6 ± 0.6	12.3 ± 0.7 *	66.3 ± 9.2	40.0 ± 6.9 *
	2	61.1 ± 6.0	64.1 ± 4.4	32.3 ± 2.0	33.0 ± 1.4	5.8 ± 0.6	5.6 ± 0.4	11.8 ± 0.8	12.5 ± 0.8 ^†^	54.3 ± 7.2	40.7 ± 7.1 *
	3	60.4 ± 5.5	62.6 ± 5.4	35.2 ± 2.8	35.8 ± 3.2	4.8 ± 0.4	4.9 ± 0.4	12.9 ± 0.7	13.7 ± 1.1 ^‡^	41.1 ± 6.0	39.0 ± 6.5
	4	65.5 ± 6.5	63.5 ± 5.6	38.4 ± 2.3	38.7 ± 2.7	4.4 ± 0.3	4.3 ± 0.4	14.5 ± 1.1	15.1 ± 0.9 ^†^	38.9 ± 5.6	35.0 ± 5.5 ^†^
	5	68.0 ± 5.3	65.3 ± 6.1	42.0 ± 2.6	41.5 ± 2.4	4.1 ± 0.5	3.9 ± 0.3	16.8 ± 1.2	16.9 ± 1.0	40.7 ± 5.0	37.6 ± 6.3
	6	80.4 ± 4.1	74.0 ± 5.8 *	53.1 ± 3.2	52.8 ± 3.8	3.6 ± 0.3	3.4 ± 0.3 ^‡^	22.6 ± 1.0	23.0 ± 1.9	41.3 ± 7.0	43.1 ± 7.6
DDFT	1	86.5 ± 7.2	83.4 ± 5.1	37.7 ± 2.2	34.8 ± 1.2 *	7.4 ± 0.3	7.8 ± 0.4 *	12.6 ± 0.9	12.8 ± 0.7	60.6 ± 8.9	50.6 ± 8.4 *
	2	74.8 ± 8.0	85.1 ± 8.3 *	33.4 ± 1.9	34.7 ± 1.7 ^†^	7.4 ± 0.6	8.2 ± 0.6 *	11.2 ± 0.7	12.2 ± 0.7 *	68.1 ± 9.4	44.8 ± 7.3 *
	3	63.9 ± 5.0	82.6 ± 9.0 *	30.1 ± 1.2	33.8 ± 1.7 *	7.4 ± 0.5	8.4 ± 0.8 *	10.1 ± 0.7	11.3 ± 0.6 *	59.5 ± 11.1	43.3 ± 7.2 *
	4	62.0 ± 7.4	80.1 ± 9.4 *	30.2 ± 1.9	33.5 ± 1.8 *	7.1 ± 0.4	8.1 ± 0.5 *	10.2 ± 0.8	11.3 ± 0.7 *	55.5 ± 10.1	46.7 ± 7.7 ^‡^
	5	95.3 ± 6.5	98.8 ± 8.5	36.7 ± 1.5	37.4 ± 1.8	8.6 ± 0.4	8.8 ± 0.5	12.9 ± 0.8	13.3 ± 0.8	54.8 ± 9.1	47.9 ± 9.4 ^†^
	6	116.9 ± 7.9	124.8 ± 7.2 *	43.7 ± 2.0	44.8 ± 1.4 ^†^	8.0 ± 0.5	8.4 ± 0.5 ^†^	17.7 ± 1.2	18.0 ± 0.8	59.8 ± 10.1	57.1 ± 9.8
ALDDFT	1	65.7 ± 5.4	24.2 ± 5.6 *	36.2 ± 1.8	24.7 ± 3.6 *	4.9 ± 0.5	2.4 ± 0.4 *	13.1 ± 0.8	10.4 ± 1.5 *	57.8 ± 10.2	50.5 ± 9.2 ^†^
	2	60.5 ± 5.0	23.4 ± 5.5 *	35.8 ± 1.9	23.1 ± 3.1 *	4.8 ± 0.4	2.5 ± 0.4 *	11.8 ± 0.9	9.6 ± 1.2 *	67.2 ± 11.3	49.8 ± 9.1 *
	3	54.5 ± 3.7	21.3 ± 4.3 *	36.8 ± 2.1	21.5 ± 2.1 *	4.4 ± 0.6	2.6 ± 0.4 *	11.3 ± 0.8	8.7 ± 1.0 *	61.7 ± 10.7	56.0 ± 10.9
	4	50.0 ± 3.2	20.1 ± 3.2 *	35.4 ± 1.9	20.7 ± 1.7 *	3.9 ± 0.2	2.5 ± 0.3 *	11.80 ± 0.7	8.5 ± 0.6 *	57.0 ± 8.6	52.0 ± 8.2 ^†^
SL	1	86.6 ± 4.3	126.3 ± 10.1 *	37.6 ± 1.9	42.8 ± 2.0 *	6.9 ± 0.6	9.3 ± 0.5 *	13.0 ± 0.7	14.9 ± 0.9 *	45.2 ± 7.0	43.2 ± 5.2
	2	84.1 ± 2.9	90.9 ± 7.1 *	36.5 ± 1.6	36.1 ± 1.5	7.0 ± 0.4	8.1 ± 0.4 *	12.4 ± 0.8	12.5 ± 0.7	55.2 ± 8.3	45.0 ± 5.7 *
	3	81.8 ± 3.2	90.9 ± 7.2 *	35.8 ± 1.8	36.3 ± 1.6	7.0 ± 0.5	8.1 ± 0.6 *	12.3 ± 0.9	12.3 ± 1.0	57.8 ± 7.4	42.1 ± 4.9 *
	4	81.8 ± 2.2	88.0 ± 5.9 *	36.0 ± 1.6	36.2 ± 1.5	6.8 ± 0.6	7.6 ± 0.6 *	12.3 ± 0.9	12.5 ± 0.9	48.7 ± 7.2	42.6 ± 5.9 ^‡^
LB-SL	1	49.4 ± 7.2	47.1 ± 4.9	26.9 ± 1.7	26.0 ± 1.4	6.6 ± 0.6	6.3 ± 0.6	8.7 ± 0.9	8.6 ± 0.7	39.9 ± 5.7	43.3 ± 7.1
	2	61.7 ± 6.5	61.0 ± 5.7	29.8 ± 2.0	29.3 ± 1.4	7.8 ± 0.9	0.8 ± 0.7	9.4 ± 1.1	9.7 ± 0.6	43.6 ± 5.6	43.9 ± 5.8
	3	101.7 ± 8.0	108.0 ± 10.1 ^†^	42.3 ± 2.0	42.8 ± 2.2	8.1 ± 0.8	0.8 ± 0.5	12.7 ± 0.9	13.6 ± 0.7 *	51.6 ± 7.6	56.6 ± 11.3
MB-SL	1	45.2 ± 5.3	43.6 ± 4.6	25.5 ± 1.7	24.8 ± 1.3	6.2 ± 0.5	6.2 ± 0.5	8.4 ± 0.9	7.9 ± 0.4 ^†^	34.8 ± 5.6	37.1 ± 5.2
	2	61.4 ± 7.3	58.4 ± 5.2	29.6 ± 1.9	28.8 ± 1.3	7.4 ± 0.6	7.1 ± 0.5	9.5 ± 1.0	9.1 ± 0.5	39.0 ± 4.7	39.2 ± 5.5
	3	100.7 ± 7.8	107.6 ± 9.9 ^†^	42.9 ± 1.9	44.0 ± 2.5	7.9 ± 0.6	7.6 ± 0.7	12.8 ± 1.0	13.7 ± 0.6 *	47.6 ± 6.0	49.9 ± 8.4

FL: forelimb; HL: hindlimb; SDFT: superficial digital flexor tendon; DDFT: deep digital flexor tendon; ALDDFT: accessory ligament of the deep digital flexor tendon; SL: suspensory ligament; LB-SL: lateral branch of the suspensory ligament; MB-SL: medial branch of the suspensory ligament; TA: transverse area; DP: dorsopalmar/dorsoplantar; LM: lateromedial. * *p* < 0.001, ^‡^
*p* < 0,01, ^†^
*p* < 0.05 indicate a significant difference between FL and HL for each variable in each zone.

**Table 2 animals-13-01411-t002:** Means and standard deviations of the morphometric variables and mean echogenicity of the digital flexor tendons and ligaments of the metacarpal palmar/plantar regions in 25 Campeiro horses.

Structure	Zone	TA (mm^2^)		Circumference (mm)		DP Length (mm)		LM Length (mm)		Mean Echogenicity	
		FL	HL	FL	HL	FL	HL	FL	HL	FL	HL
SDFT	1	70.3 ± 6.6	67.5 ± 4.7	32.8 ± 1.7	33.7 ± 1.9	6.6 ± 0.6	5.9 ± 0.4 *	12.2 ± 0.7	13.2 ± 1.0 *	61.1 ± 8.9	39.4 ± 6.4 *
	2	65.1 ± 6.4	66.8 ± 4.2	32.9 ± 2.0	33.9 ± 1.6	6.0 ± 0.7	5.7 ± 0.4	12.2 ± 0.8	13.3 ± 0.9 *	50.0 ± 9.8	40.9 ± 7.8 *
	3	63.0 ± 6.0	69.4 ± 4.7 *	35.1 ± 2.4	37.4 ± 2.2^‡^	4.9 ± 0.6	4.9 ± 0.4	13.6 ± 1.1	14.7 ± 1.0 ^‡^	41.1 ± 6.6	40.1 ± 6.9
	4	67.6 ± 4.7	68.6 ± 5.1	38.6 ± 2.0	39.7 ± 1.8	4.6 ± 0.3	4.4 ± 0.3	15.2 ± 0.9	15.9 ± 0.7 ^‡^	38.1 ± 5.5	35.4 ± 6.7
	5	72.1 ± 4.7	70.5 ± 5.6	43.0 ± 1.9	42.4 ± 2.0	4.3 ± 0.2	4.0 ± 0.3 ^‡^	17.7 ± 0.8	17.7 ± 1.0	38.9 ± 7.2	38.8 ± 7.2
	6	83.7 ± 3.6	82.3 ± 4.6	55.3 ± 2.4	55.0 ± 3.0	3.6 ± 0.2	3.7 ± 0.3	23.7 ± 1.3	24.0 ± 1.4	37.9 ± 5.2	41.2 ± 7.0
DDFT	1	86.9 ± 6.5	89.2 ± 6.9	36.9 ± 2.3	36.1 ± 1.5	7.9 ± 0.4	8.1 ± 0.4 ^†^	12.8 ± 1.0	13.5 ± 0.8 ^†^	58.9 ± 8.8	51.0 ± 8.5 ^‡^
	2	75.4 ± 6.7	86.6 ± 5.7 *	32.9 ± 1.8	35.0 ± 1.3 *	7.9 ± 0.7	8.3 ± 0.4 ^†^	11.3 ± 0.8	12.7 ± 0.7 *	63.8 ± 9.3	46.3 ± 10.1 *
	3	65.4 ± 5.6	85.5 ± 4.5 *	30.1 ± 2.0	34.6 ± 1.0 *	8.0 ± 0.4	8.4 ± 0.5 ^‡^	10.0 ± 0.7	12.1 ± 0.7 *	54.7 ± 8.7	41.9 ± 8.0 *
	4	63.7 ± 4.9	83.8 ± 5.6 *	29.5 ± 1.2	34.2 ± 1.2 *	7.6 ± 0.4	8.3 ± 0.3 *	10.1 ± 0.5	12.1 ± 0.6 *	50.5 ± 10.1	45.0 ± 8.5 ^†^
	5	98.8 ± 7.1	99.8 ± 8.6	37.2 ± 1.4	37.6 ± 1.7	8.8 ± 0.6	8.7 ± 0.6	13.3 ± 0.7	13.5 ± 0.9	48.0 ± 10.0	49.1 ± 8.8
	6	124.1 ± 9.0	128.1 ± 8.6	44.9 ± 2.4	45.8 ± 2.0	8.4 ± 0.4	8.4 ± 0.4	18.3 ± 1.3	18.6 ± 1.1	50.7 ± 9.0	54.8 ± 10.2
ALDDFT	1	65.7 ± 6.9	28.9 ± 7.2 *	35.9 ± 2.1	27.9 ± 4.1 *	5.0 ± 0.6	2.6 ± 0.5 *	13.5 ± 1.2	11.5 ± 1.8 *	56.6 ± 9.0	56.8 ± 9.8
	2	61.1 ± 6.0	25.1 ± 4.2 *	35.0 ± 1.8	24.8 ± 2.9 *	4.8 ± 0.5	2.5 ± 0.5 *	12.6 ± 0.8	10.3 ± 1.4 *	60.4 ± 10.8	59.4 ± 8.5
	3	54.8 ± 5.5	22.1 ± 3.3 *	35.9 ± 2.3	23.2 ± 2.5 *	4.3 ± 0.4	2.4 ± 0.2 *	12.4 ± 0.8	9.5 ± 1.1 *	55.4 ± 8.6	59.4 ± 10.8
	4	52.6 ± 5.8	19.7 ± 2.9 *	36.5 ± 1.8	21.8 ± 1.7 *	4.0 ± 0.4	2.3 ± 0.3 *	12.6 ± 0.6	8.9 ± 0.8 *	54.4 ± 6.7	54.6 ± 9.3
SL	1	90.8 ± 6.6	135.8 ± 8.7 *	37.6 ± 1.6	44.4 ± 1.9 *	7.3 ± 0.5	9.6 ± 0.4 *	13.5 ± 0.7	15.9 ± 0.9 *	47.8 ± 6.2	43.9 ± 6.0 ^†^
	2	87.3 ± 4.5	87.7 ± 5.9	36.1 ± 1.0	35.6 ± 1.4	7.4 ± 0.4	7.8 ± 0.4 ^‡^	12.7 ± 0.6	12.2 ± 0.6 ^†^	52.8 ± 9.9	45.3 ± 6.4 ^‡^
	3	85.0 ± 3.9	87.3 ± 5.2	35.8 ± 1.2	35.8 ± 1.3	7.3 ± 0.4	7.6 ± 0.3 ^‡^	12.5 ± 0.7	12.6 ± 0.7	55.1 ± 7.2	45.5 ± 5.2 *
	4	84.9 ± 3.2	87.1 ± 4.6	35.5 ± 0.8	36.0 ± 1.2	7.3 ± 0.3	7.5 ± 0.3	12.5 ± 0.6	12.4 ± 0.6	50.7 ± 9.0	44.6 ± 5.0 ^‡^
LB-SL	1	46.9 ± 3.2	46.7 ± 5.2	25.3 ± 0.9	25.5 ± 1.3	6.7 ± 0.4	6.6 ± 0.5	8.5 ± 0.4	8.3 ± 0.6	38.4 ± 6.4	43.8 ± 6.2 ^‡^
	2	59.5 ± 3.8	60.4 ± 5.0	28.5 ± 1.0	28.8 ± 1.3	7.5 ± 0.3	7.8 ± 0.6 ^†^	9.5 ± 0.5	9.5 ± 0.7	40.7 ± 7.7	42.2 ± 6.5
	3	103.5 ± 7.6	109.1 ± 10.1 ^†^	40.8 ± 1.8	42.1 ± 2.1 ^†^	8.2 ± 0.4	8.4 ± 0.5 ^†^	13.1 ± 0.6	13.7 ± 0.7 ^‡^	46.7 ± 8.9	53.8 ± 11.0 ^†^
MB-SL	1	44.2 ± 4.0	43.7 ± 4.3	24.7 ± 1.1	24.8 ± 1.5	6.3 ± 0.4	6.3 ± 0.7	8.4 ± 0.4	8.3 ± 0.5	33.1 ± 4.6	39.2 ± 6.7 *
	2	57.2 ± 5.0	57.4 ± 5.8	28.2 ± 1.3	28.1 ± 1.3	7.2 ± 0.4	7.1 ± 0.7	9.5 ± 0.6	9.3 ± 0.6	39.8 ± 5.7	39.2 ± 6.7
	3	105.2 ± 8.5	109.4 ± 6.9	42.0 ± 1.9	42.7 ± 1.4	8.0 ± 0.4	8.3 ± 0.5	13.0 ± 0.6	13.3 ± 0.6 ^†^	47.6 ± 7.3	45.9 ± 9.1

FL: forelimb; HL: hindlimb; SDFT: superficial digital flexor tendon; DDFT: deep digital flexor tendon; ALDDFT: accessory ligament of the deep digital flexor tendon; SL: suspensory ligament; LB-SL: lateral branch of the suspensory ligament; MB-SL: medial branch of the suspensory ligament; TA: transverse area; DP: dorsopalmar/dorsoplantar; LM: lateromedial * *p* < 0.001, ^‡^
*p* < 0.01, ^†^
*p* < 0.05 indicate a significant difference between FL and HL for each variable in each zone.

**Table 3 animals-13-01411-t003:** Mean difference between the zones of the digital flexor tendons and ligaments of the palmar metacarpal region in 25 Mangalarga Marchador horses.

Structure	Zone	Zone′	TA (mm^2^)	C (mm)	DP (mm)	LM (mm)	ME
			Mean Difference (Zone–Zone′)				
SDFT	1	2	4.54 ^†^	-	0.69	-	12.00
		3	5.24 ^‡^	−3.06	1.63	−1.30	25.20
		4	-	−6.22	2.05	−2.86	27.41
		5	-	−9.89	2.34	−5.23	25.55
		6	−14.79	−20.99	2.81	−11.04	24.96
	2	3	-	−2.88	0.95	−1.06	13.21
		4	−4.43 ^†^	−6.04	1.36	−2.61	15.41
		5	−6.94	−9.71	1.65	−4.99	13.55
		6	−19.34	−20.81	2.12	−10.80	12.96
	3	4	−5.12 ^‡^	−3.16	0.41 ^‡^	−1.56	-
		5	−7.63	−6.83	0.71	−3.94	-
		6	−20.03	−17.93	1.18	−9.75	-
	4	5	-	−3.67	-	−2.38	-
		6	−14.91	−14.78	0.76	−8.19	-
	5	6	−12.40	−11.10	0.47 ^‡^	−5.81	-
DDFT	1	2	11.71	4.38	-	1.40	−7.45 ^†^
		3	22.59	7.67	-	2.53	-
		4	24.49	7.50	-	2.43	-
		5	−8.84	-	−1.19	-	-
		6	−30.39	−5.99	−0.65	−5.06	-
	2	3	10.88	3.29	-	1.12	8.59 ^‡^
		4	12.78	3.12	-	1.02	12.54
		5	−20.54	−3.30	−1.15	−1.67	13.33
		6	−42.10	−10.37	−0.60	−6.46	8.31 ^‡^
	3	4	-	-	-	-	-
		5	−31.43	−6.59	−1.14	−2.79	-
		6	−52.98	−13.66	−0.60	−7.58	-
	4	5	−33.32	−6.43	−1.43	−2.69	-
		6	−54.88	−13.49	−0.88	−7.49	-
	5	6	−21.55	−7.07	0.55	−4.79	-
ALDDFT	1	2	5.12 ^‡^	-	-	1.35	−9.40
		3	11.19	-	0.52	1.82	-
		4	15.68	-	0.97	1.30	-
	2	3	60.72	-	0.43 ^‡^	-	-
		4	10.56	-	0.87	-	10.11
	3	4	44.92 ^‡^	-	0.45 ^‡^	-	-
SL	1	2	-	-	-	-	−10.00
		3	4.78 ^‡^	-	-	0.65 ^†^	−12.57
		4	4.77 ^‡^	-	-	-	-
	2	3	-	-	-	-	-
		4	-	-	-	-	6.43 ^†^
	3	4	-	-	-	-	9.01
LB-SL	1	2	−12.29	−2.95	−0.71 ^†^	−1.22	-
		3	−52.30	−15.40	−4.03	−1.48	−11.69
	2	3	−40.02	−12.45	−3.33	-	−8.00
MB-SL	1	2	−16.18	−4.11	−1.06	−1.19	−4.21 ^†^
		3	−55.48	−17.32	−4.41	−1.76	−12.73
	2	3	−39.30	−13.21	−3.35	−0.57 ^‡^	−8.52

SDFT: superficial digital flexor tendon; DDFT: deep digital flexor tendon; ALDDFT: accessory ligament of the deep digital flexor tendon; SL: suspensory ligament; LB-SL: lateral branch of the suspensory ligament; MB-SL: medial branch of the suspensory ligament; TA: transverse area; C: circumference; DP: dorsopalmar length; LM: lateromedial length; ME: mean echogenicity. ^†^
*p* < 0.05; ^‡^
*p* < 0.01; nothing marked *p* < 0.001; only significant differences are demonstrated.

**Table 4 animals-13-01411-t004:** Mean difference between the zones of the digital flexor tendons and ligaments of the palmar metacarpal region in 25 Campeiro horses.

Structure	Zone	Zone′	TA (mm^2^)	C (mm)	DP (mm)	LM (mm)	ME
			Mean Difference (Zone–Zone′)				
SDFT	1	2	5.23 ^‡^	-	0.67	-	11.04
		3	7.30	−2.26	1.71	−1.45	19.94
		4	-	−5.80	2.04	−3.04	23.02
		5	-	−10.14	2.38	−5.58	22.17
		6	−13.40	−22.46	3.03	−11.55	23.18
	2	3	-	−2.21	1.05	−1.45	8.90
		4	-	−5.75	1.37	−3.04	11.97
		5	−7.03	−10.09	1.72	−5.58	11.13
		6	−18.63	−22.41	2.36	−11.56	12.14
	3	4	−4.59 ^†^	−3.54	-	−1.59	-
		5	−9.10	−7.88	0.67	−4.13	-
		6	−20.70	−20.20	1.31	−10.11	-
	4	5	−4.52 ^†^	−4.34	0.34 ^†^	−2.54	-
		6	−16.11	−16.66	0.99	−8.52	-
	5	6	−11.60	−12.33	0.64	−5.98	-
DDFT	1	2	11.48	4.03	-	1.58	-
		3	21.42	6.77	-	2.81	-
		4	23.16	7.39	-	2.77	8.45
		5	−11.89	-	−0.92	-	10.94
		6	−37.23	−8.04	−4.72	−5.45	8.27 ^‡^
	2	3	9.95	2.74	-	1.23	9.18
		4	11.68	3.36	-	1.19	13.34
		5	−23.37	−4.36	−0.92	−2.06	15.83
		6	−48.70	−12.06	−0.47	−7.03	13.16
	3	4	-	-	0.37 ^†^	-	-
		5	−33.32	−7.10	−0.85	−3.30	6.65 ^†^
		6	−58.65	−14.80	−0.39 ^‡^	−8.26	-
	4	5	−35.05	−7.71	−1.22	−3.26	-
		6	−60.38	−15.42	−0.77	−8.22	-
	5	6	−25.34	−7.71	0.45 ^‡^	−4.96	-
ALDDFT	1	2	4.63 ^†^	-	-	0.85 ^‡^	-
		3	10.94	-	0.65	1.11	-
		4	13.14	-	1.03	0.89	-
	2	3	6.30	-	0.41 ^‡^	-	-
		4	8.51	−1.50 ^†^	0.80	-	-
	3	4	-	-	0.38 ^†^	-	-
SL	1	2	-	1.47 ^†^	-	0.79 ^‡^	-
		3	5.78 ^‡^	1.82 ^‡^	-	0.96	−7.28 ^†^
		4	5.93	2.08	-	1.02	-
	2	3	-	-	-	-	-
		4	-	-	-	-	-
	3	4	-	-	-	-	-
LB-SL	1	2	−12.58	−3.23	−0.79	−1.03	-
		3	−56.58	−15.53	−1.48	−4.57	−8.27
	2	3	−44.01	−12.30	−0.69	−3.54	−5.99 ^‡^
MB-SL	1	2	−13.01	−3.46	−0.87	−1.10	−6.72 ^‡^
		3	−60.68	−17.25	−1.71	−4.58	−14.46
	2	3	−47.98	−13.79	−0.84	−3.48	−7.75

SDFT: superficial digital flexor tendon; DDFT: deep digital flexor tendon; ALDDFT: accessory ligament of the deep digital flexor tendon; SL: suspensory ligament; LB-SL: lateral branch of the suspensory ligament; MB-SL: medial branch of the suspensory ligament; TA: transverse area; C: circumference; DP: dorsopalmar length; LM: lateromedial length; ME: mean echogenicity. ^†^
*p* < 0.05; ^‡^
*p* < 0.01; nothing marked *p* < 0.001; only significant differences are demonstrated.

**Table 5 animals-13-01411-t005:** Mean difference between the zones of the digital flexor tendon and ligament of the metatarsal plantar region in 25 Mangalarga Marchador horses.

Structure	Zone	Zone′	TA (mm^2^)	C (mm)	DP (mm)	LM (mm)	ME
			Mean Difference (Zone–Zone′)				
SDFT	1	2	-	-	-	-	-
		3	-	−3.57	0.76	−1.40	-
		4	-	−6.56	1.33	−2.83	-
		5	-	−9.29	1.76	−4.64	-
		6	−11.47	−20.66	2.24	−10.69	-
	2	3	-	−2.74	0.66	−1.21	-
		4	-	−5.73	1.23	−2.64	5.72 ^†^
		5	-	−8.46	1.66	−4.45	-
		6	−9.88	−19.83	2.14	−10.49	-
	3	4	-	−2.99	0.57	−1.43	-
		5	-	−5.72	0.99	−3.24	-
		6	−11.44	−17.08	1.48	−9.29	-
	4	5	-	−2.73	0.43 ^‡^	−1.81	-
		6	−10.48	−14.10	0.91	−7.86	−8.16
	5	6	−8.69	−11.36	0.48	−6.04	-
DDFT	1	2	-	-	−0.39 ^‡^	-	5.83 ^†^
		3	-	-	−0.66	1.41	7.32 ^‡^
		4	-	-	−0.33 ^†^	1.47	-
		5	−15.36	−2.58	−1.00	-	-
		6	−41.38	−9.98	−0.60	−5.24	−6.51^‡^
	2	3	-	-	-	0.87^‡^	-
		4	4.98 ^†^	-	-	0.93	-
		5	−13.66	−2.75	−0.61	−1.04	-
		6	−39.67	−10.15	-	−5.77	−12.33
	3	4	-	-	0.33 ^†^	-	-
		5	−16.19	−3.64	−0.33 ^†^	−1.91	-
		6	−42.20	−11.04	-	−6.65	−13.83
	4	5	−18.64	−3.91	−0.66	−1.97	-
		6	−44.65	−11.31	-	−6.71	−10.42
	5	6	−26.01	−7.40	0.39 ^‡^	−4.74	−9.20
ALDDFT	1	2	-	-	-	-	-
		3	-	3.15	-	1.67	-
		4	-	3.87	-	1.90	-
	2	3	-	-	-	0.92 ^‡^	−6.22 ^†^
		4	-	4.17	-	1.15	-
	3	4	-		-	-	-
SL	1	2	35.36	6.69	1.20	2.41	-
		3	35.39	6.52	1.23	2.59	-
		4	38.29	6.53	1.69	2.44	-
	2	3	-	-	-	-	-
		4	-	-	0.50	-	-
	3	4	-	-	0.46	-	-
LB-SL	1	2	−13.88	−3.39	−1.25	−1.11	-
		3	−60.87	−16.87	−1.65	−4.96	−13.34
	2	3	−46.99	−13.48	−0.39 ^†^	−3.85	−12.70
MB-SL	1	2	−14.87	−4.06	−0.95	−1.23	-
		3	−64.04	−19.26	−1.47	−5.78	−12.73
	2	3	−49.17	−15.20	−0.53 ^‡^	−4.56	−10.65

SDFT: superficial digital flexor tendon; DDFT: deep digital flexor tendon; ALDDFT: accessory ligament of the deep digital flexor tendon; SL: suspensory ligament; LB-SL: lateral branch of the suspensory ligament; MB-SL: medial branch of the suspensory ligament; TA: transverse area; C: circumference; DP: dorsoplantar length; LM: lateromedial length; ME: mean echogenicity. ^†^
*p* < 0.05; ^‡^
*p* < 0.01; nothing marked *p* < 0.001; only significant differences are demonstrated.

**Table 6 animals-13-01411-t006:** Mean difference between the zones of the digital flexor tendons and ligaments of the metatarsal plantar region in 25 Campeiro horses.

Structure	Zone	Zone′	TA (mm^2^)	C (mm)	DP (mm)	LM (mm)	ME
			Mean Difference (Zone–Zone′)				
SDFT	1	2	-	-	-	-	-
		3	-	−3.72	0.97	−1.45	-
		4	-	−6.06	1.47	−2.67	-
		5	-	−8.75	1.92	−4.46	-
		6	−14.81	−21.36	2.25	−10.78	-
	2	3	-	−3.52	0.73	−1.37	-
		4	-	−5.85	1.24	−2.59	-
		5	-	−8.54	1.68	−4.37	-
		6	−15.50	−21.15	2.02	−10.70	-
	3	4	-	−2.33	0.50	−1.22	-
		5	-	−5.03	0.95	−3.00	-
		6	−12.84	−17.64	1.28	−9.33	-
	4	5	-	−2.69	0.45	−1.79	-
		6	−13.63	−15.30	0.78	−8.11	-
	5	6	−11.88	−12.61	0.33^‡^	−6.33	-
DDFT	1	2	-	-	-	0.84 ^‡^	-
		3	-	1.51 ^†^	-	1.37	9.11
		4	5.39 ^‡^	1.86	-	1.42	6.08 ^†^
		5	−10.56	−1.52 ^†^	−0.55	-	-
		6	−38.89	−9.74	−0.26 ^†^	−5.12	-
	2	3	-	-	-	-	-
		4	-	-	-	-	-
		5	−13.16	−2.63	−0.39	−0.87 ^‡^	-
		6	−41.49	−10.86	-	−5.96	−8.47
	3	4	-	-	-	-	-
		5	−14.24	−3.03	−0.32 ^‡^	−1.41	−7.14 ^‡^
		6	−42.56	−11.25	-	−6.50	−12.82
	4	5	−15.96	−3.38	−0.42	−1.45	-
		6	−44.28	−11.60	-	−6.54	−9.79
	5	6	−28.32	−8.22	0.29^†^	−5.09	-
ALDDFT	1	2	-	3.13	-	1.25	-
		3	6.70	4.70	-	1.97	-
		4	9.23	6.09	0.34 ^†^	2.60	-
	2	3	-	-	-	-	-
		4	5.35 ^‡^	2.95	-	1.34	-
	3	4	-	-	-	-	-
SL	1	2	48.18	8.81	1.73	3.69	-
		3	48.51	8.59	1.93	3.33	-
		4	48.77	8.42	2.08	3.47	-
	2	3	-	-	-	-	-
		4	-	-	0.35 ^‡^	-	-
	3	4	-	-	-	-	-
LB-SL	1	2	−13.68	−3.32	−1.12	−1.22	-
		3	−62.36	−16.63	−1.81	−5.37	−10.01
	2	3	−48.67	−13.32	−0.69	−4.16	−11.56
MB-SL	1	2	−13.63	−3.36	−0.87	−1.01	-
		3	−65.59	−17.95	−1.99	−5.08	−6.68 ^‡^
	2	3	−51.96	−14.59	−1.11	−4.07	−6.70 ^‡^

SDFT: superficial digital flexor tendon; DDFT: deep digital flexor tendon; ALDDFT: accessory ligament of the deep digital flexor tendon; SL: suspensory ligament; LB-SL: lateral branch of the suspensory ligament; MB-SL: medial branch of the suspensory ligament; TA: transverse area; C: circumference; DP: dorsoplantar length; LM: lateromedial length; ME: mean echogenicity. ^†^
*p* < 0.05; ^‡^
*p* < 0.01; nothing marked *p* < 0.001; only significant differences are demonstrated.

**Table 7 animals-13-01411-t007:** Mean difference between digital flexor tendons and ligaments in each zone of the palmar metacarpal region in 25 Mangalarga Marchador horses.

Zone	Structure	Structure′	TA (mm^2^)	C (mm)	DP (mm)	LM (mm)	ME
			Mean Difference (Structure–Structure′)				
1	SDFT	DDFT	−20.84	−5.60	−0.92	−1.04	-
		ALDDFT	-	−4.04	1.56	−1.52	8.509
		SL	−20.97	−5.45	−0.40 ^‡^	−1.37	21.08
	DDFT	ALDDFT	20.82	1.56 ^†^	2.48	-	-
		SL	-	-	0.52	-	15.45
	LATFP	SL	−20.95	−1.40 ^†^	−1.96	-	12.58
	LB-SL	MB-SL	-	1.33 ^†^	0.44 ^†^	-	5.12 ^‡^
2	SDFT	DDFT	−13.68	-	−1.65	0.60 ^†^	−13.81
		ALDDFT	-	−3.48	0.96	-	−12.88
		SL	−22.97	−4.15	−1.25	−0.59 ^†^	-
	DDFT	ALDDFT	14.23	−2.44	2.61	-	-
		SL	−9.29	−3.11	0.40 ^‡^	−1.19	12.90
	ALDDFT	SL	−23.52	-	−2.21	−0.66 ^†^	11.97
	LB-SL	MB-SL	-	-	0.47 ^†^	-	4.61 ^†^
3	SDFT	DDFT	-	5.13	−2.60	2.78	−18.44
		ALDDFT	5.93	−1.58 ^†^	0.44 ^‡^	1.60	−20.69
		SL	−21.42	-	−2.15	0.57 ^†^	−16.69
	DDFT	ALDDFT	9.42	−6.71	3.05	−1.18	-
		SL	−17.93	−5.74	0.45	−2.21	-
	ALDDFT	SL	−27.35	-	−2.59	−1.03	-
	LB-SL	MB-SL	-	-	-	-	4.09 ^†^
4	SDFT	DDFT	-	8.12	−2.73	4.24	−16.69
		ALDDFT	15.54	2.99	0.48	2.64	−18.19
		SL	−16.31	2.35	−2.42	2.10	−9.89
	DDFT	ALDDFT	12.02	−5.13	3.21	−1.60	-
		SL	−19.84	−5.77	-	−2.14	9.80 ^†^
	ALDDFT	SL	−31.86	-	−2.90	-	8.29
5	SDFT	DDFT	−27.29	5.37	−4.45	3.93	−14.03
6	SDFT	DDFT	−36.44	9.40	−4.38	4.94	−18.47

SDFT: superficial digital flexor tendon; DDFT: deep digital flexor tendon; ALDDFT: accessory ligament of the deep digital flexor tendon; SL: suspensory ligament; LB-SL: lateral branch of the suspensory ligament; MB-SL: medial branch of the suspensory ligament; TA: transverse area; C: circumference; DP: dorsopalmar length; LM: lateromedial length; ME: mean echogenicity. ^†^
*p* < 0.05; ^‡^
*p* < 0.01; nothing marked *p* < 0.001; only significant differences are demonstrated.

**Table 8 animals-13-01411-t008:** Mean difference between the digital flexor tendons and ligaments in each zone of the metacarpal palmar region in 25 Campeiro horses.

Zone	Structure	Structure′	TA (mm^2^)	C (mm)	DP (mm)	LM (mm)	ME
			Mean Difference (Structure–Structure′)				
1	SDFT	DDFT	−16.57	−4.05	−1.24	−0.66 ^†^	-
		ALDDFT	4.58 ^†^	−3.11	1.65	−1.30	-
		SL	−20.51	−4.79	−0.65	−1.31	13.24
	DDFT	ALDDFT	21.14	-	2.89	−0.64 ^†^	-
		SL	−3.94 ^†^	-	0.59	−0.65 ^†^	11.11
	LATFP	SL	−25.09	−1.68 ^‡^	−2.30	-	8.73
	LB-SL	MB-SL	-	-	0.38 ^‡^	-	5.34 ^†^
2	SDFT	DDFT	−10.32	-	−1.92	0.91	−13.81
		ALDDFT	3.98 ^†^	−2.08	1.21	-	−10.39
		SL	−22.23	−3.27	−1.45	-	-
	DDFT	ALDDFT	14.30	−2.10	3.13	−1.37	-
		SL	−11.90	−3.29	0.47	−1.43	11.09
	ALDDFT	SL	−26.20	-	−2.66	-	7.66 ^‡^
	LB-SL	MB-SL	-	-	0.31^†^	-	-
3	SDFT	DDFT	-	4.98	−3.04	3.60	−13.52
		ALDDFT	8.21	-	0.58	1.25	−14.29
		SL	−22.03	-	−2.33	1.10	−13.98
	DDFT	ALDDFT	10.66	−5.76	3.62	−2.35	-
		SL	−19.58	−5.69	0.70	−2.50	-
	ALDDFT	SL	−30.24	-	−2.91	-	-
	LB-SL	MB-SL	-	−1.14 ^‡^	-	-	-
4	SDFT	DDFT	-	9.14	−2.99	5.15	−12.45
		ALDDFT	15.00	2.17	0.64	2.62	−16.32
		SL	−17.30	3.09	−2.72	2.75	−12.66
	DDFT	ALDDFT	11.13	−6.97	3.63	−2.53	-
		SL	−21.17	−6.05	-	−2.40	-
	ALDDFT	SL	−32.30	-	−3.36	-	-
5	SDFT	DDFT	−26.66	5.76	−4.55	4.43	−9.10
6	SDFT	DDFT	−40.40	10.38	−4.74	5.44	−12.78

SDFT: superficial digital flexor tendon; DDFT: deep digital flexor tendon; ALDDFT: accessory ligament of the deep digital flexor tendon; SL: suspensory ligament; LB-SL: lateral branch of the suspensory ligament; MB-SL: medial branch of the suspensory ligament; TA: transverse area; C: circumference; DP: dorsopalmar length; LM: lateromedial length; ME: mean echogenicity. ^†^
*p* < 0.05; ^‡^
*p* < 0.01; nothing marked *p* < 0.001; only significant differences are demonstrated.

**Table 9 animals-13-01411-t009:** Mean difference between digital flexor tendons and ligaments in each zone of the metatarsal plantar region in 25 Mangalarga Marchador horses.

Zone	Structure	Structure′	TA (mm^2^)	C (mm)	DP (mm)	LM (mm)	ME
			Mean Difference (Structure–Structure′)				
1	SDFT	DDFT	−20.89	−2.66	−2.11	-	−10.62
		ALDDFT	38.36	7.52	3.28	1.88	−10.55
		SL	−63.75	−10.59	−3.62	−2.63	-
	DDFT	ALDDFT	59.26	10.18	5.39	2.37	-
		SL	−42.86	−7.92	−1.51	−2.14	7.37
	LATFP	SL	−102.10	−18.10	−6.90	−4.50	7.30
	LB-SL	MB-SL	-	1.17 ^†^	-	0.69	6.13 ^‡^
2	SDFT	DDFT	−21.00	−1.66 ^‡^	−2.60	-	-
		ALDDFT	40.71	9.87	3.04	2.82	−9.09
		SL	−26.80	−3.06	−2.53	-	-
	DDFT	ALDDFT	61.71	11.53	5.64	2.58	-
		SL	−5.80 ^‡^	−1.40 ^†^	-	-	-
	ALDDFT	SL	−67.51	−12.93	−5.56	−2.85	-
	LB-SL	MB-SL	-		0.44 ^†^	0.57 ^‡^	-
3	SDFT	DDFT	−20.03	1.97 ^‡^	−3.54	2.32	-
		ALDDFT	41.23	14.24	2.32	4.95	−16.99
		SL	−28.33	-	−3.15	1.36	-
	DDFT	ALDDFT	61.26	12.27	5.86	2.63	−12.75
		SL	−8.30	−2.47	0.39 ^‡^	−0.96	-
	ALDDFT	SL	−69.56	−14.74	−5.47	−3.58	13.87
	LB-SL	MB-SL	-	−1.22	-	-	6.73 ^‡^
4	SDFT	DDFT	−16.62	5.22	−3.78	3.81	−11.69
		ALDDFT	43.44	17.95	1.80	6.60	−17.05
		SL	−24.46	2.51	−3.26	2.64	−7.59
	DDFT	ALDDFT	60.06	12.72	5.58	2.79	−5.36 ^†^
		SL	−7.84	−2.72	0.52	−1.17	-
	ALDDFT	SL	−67.90	−15.44	−5.06	−3.96	9.46
5	SDFT	DDFT	−33.48	4.05	−4.87	3.65	−10.25
6	SDFT	DDFT	−50.80	8.01	−4.96	4.96	−13.95

SDFT: superficial digital flexor tendon; DDFT: deep digital flexor tendon; ALDDFT: accessory ligament of the deep digital flexor tendon; SL: suspensory ligament; LB-SL: lateral branch of the suspensory ligament; MB-SL: medial branch of the suspensory ligament; TA: transverse area; C: circumference; DP: dorsopalmar length; LM: lateromedial length; ME: mean echogenicity. ^†^
*p* < 0.05; ^‡^
*p* < 0.01; nothing marked *p* < 0.001; only significant differences are demonstrated.

**Table 10 animals-13-01411-t010:** Mean difference between the digital flexor tendons and ligaments in each zone of the metatarsal plantar region in 25 Campeiro horses.

Zone	Structure	Structure′	TA (mm^2^)	C (mm)	DP (mm)	LM (mm)	ME
			Mean Difference (Structure–Structure′)				
1	SDFT	DDFT	−21.75	−2.42	−2.23	-	−11.64
		ALDDFT	3851	5.77	3.30	1.70	−17.42
		SL	−68.38	−10.73	−3.66	−2.68	-
	DDFT	ALDDFT	60.26	8.19	5.52	2.00	−5.78 ^†^
		SL	−46.63	−8.31	−1.43	−2.38	7.15 ^‡^
	LATFP	SL	−106.90	−16.50	−6.95	−4.38	12.93
	LB-SL	MB-SL	-	-	-	-	-
2	SDFT	DDFT	−19.83	-	−2.62	0.62^†^	−5.43 ^†^
		ALDDFT	41.71	9.11	3.17	3.03	−18.51
		SL	−20.88	−1.72	−2.16	1.09	-
	DDFT	ALDDFT	61.54	10.21	5.79	2.41	−13.08
		SL	-	-	0.46	-	-
	ALDDFT	SL	−62.59	−10.83	−5.33	−1.94	14.11
	LB-SL	MB-SL	-	-	0.61	-	-
3	SDFT	DDFT	−16.10	2.82	−3.43	2.53	-
		ALDDFT	47.28	14.2	2.56	5.12	−19.29
		SL	−17.90	1.58 ^‡^	−2.70	2.10	−5.39 ^†^
	DDFT	ALDDFT	63.38	11.38	5.99	2.59	−17.43
		SL	-	−1.24 ^†^	0.73	-	-
	ALDDFT	SL	−65.18	−12.61	−5.26	−3.01	13.90
	LB-SL	MB-SL	-	-	-	-	7.88
4	SDFT	DDFT	−15.18	5.50	−3.82	3.79	−9.60
		ALDDFT	48.92	17.91	2.16	6.96	−19.24
		SL	−18.43	3.74	−3.05	3.46	−9.23
	DDFT	ALDDFT	64.09	12.41	5.99	3.17	−9.64
		SL	-	−1.75	0.78	-	-
	ALDDFT	SL	−67.35	−14.17	−5.21	−3.50	10.01
5	SDFT	DDFT	−29.38	4.81	−4.70	4.12	−10.25
6	SDFT	DDFT	−45.82	9.20	−4.74	5.36	−13.54

SDFT: superficial digital flexor tendon; DDFT: deep digital flexor tendon; ALDDFT: accessory ligament of the deep digital flexor tendon; SL: suspensory ligament; LB-SL: lateral branch of the suspensory ligament; MB-SL: medial branch of the suspensory ligament; TA: transverse area; C: circumference; DP: dorsopalmar length; LM: lateromedial length; ME: mean echogenicity. ^†^
*p* < 0.05; ^‡^
*p* < 0.01; nothing marked *p* < 0.001; only significant differences are demonstrated.

**Table 11 animals-13-01411-t011:** Mean difference in morphometric variables and mean echogenicity of palmar/plantar metacarpal/metatarsal structures between the Mangalarga Marchador (*n* = 25) and Campeiro (*n* = 25) breeds.

Structure	Zone	TA (mm^2^) *		C (mm) *		DP (mm) *		LM (mm) *		ME (mm) *	
		FL	HL	FL	HL	FL	HL	FL	HL	FL	HL
SDFT	1	−4.67 ^†^	−4.94	-	−1.48 ^‡^	-	−0.24 ^†^	−0.58 ^‡^	−0.94	-	-
	2	−3.98 ^†^	−2.67 ^†^	-	-	-	-	-	−0.84 ^‡^	-	-
	3	-	−6.88	-	-	-	-	−0.73 ^‡^	−1.00 ^‡^	-	-
	4	-	−5.13 ^‡^	-	-	−0.21 ^‡^	-	−0.76 ^†^	−0.79 ^‡^	-	-
	5	−4.08 ^‡^	−5.17 ^‡^	-	-	-	-	−0.92 ^‡^	−0.76 ^†^	-	-
	6	−3.28 ^‡^	−8.28	−2.15 ^†^	−2.18 ^†^	-	−0.23 ^‡^	−1.09 ^‡^	−1.04 ^†^	-	-
DDFT	1	-	−5.80 ^‡^	-	−1.23 ^‡^	−0.52	−0.37 ^‡^	-	−0.76 ^‡^	-	-
	2	-	-	-	-	−0.48 ^†^	-	-	−0.46 ^†^	-	-
	3	-	-	-	-	−0.55	-	-	−0.79	-	-
	4	-	-	-	-	−0.46	-	-	−0.81	-	-
	5	-	-	-	-	-	-	-	-	6.75 ^†^	-
	6	−7.24 ^‡^	-	-	-	−0.35 ^‡^	-	-	−0.64 ^†^	9.10 ^‡^	-
ALDDFT	1	-	−4.80 ^†^	-	−3.22 ^‡^	-	-	-	−1.13 ^†^	-	−6.30 ^†^
	2	-	-	-	-	-	-	−0.86 ^‡^	-	6.74 ^†^	−9.57
	3	-	-	-	−1.67 ^†^	-	-	−1.08	−0.83 ^†^	6.32 ^†^	-
	4	-	-	−1.09 ^†^	-	-	-	−0.78	-	-	-
SL	1	−4.22 ^†^	−9.59 ^‡^	-	−1.62 ^‡^	−0.45 ^‡^	−0.28 ^†^	−0.51 ^†^	−0.10	-	-
	2	−3.24 ^‡^	-	-	-	−0.41 ^‡^	0.26 ^†^	-	-	-	-
	3	−3.22 ^‡^	-	-	-	−0.30 ^†^	0.42 ^‡^	-	-	-	−3.31 ^†^
	4	−3.06	-	-	-	−0.50	-	-	-	-	-
LB-SL	1	-	-	1.57	-	-	−0.32 ^†^	-	-	-	-
	2	-	-	1.28 ^‡^	-	0.37 ^†^	-	-	-	-	-
	3	-	-	1.43 ^†^	-	-	−0.49	-	-	4.94 ^†^	-
MB-SL	1	-	-	-	-	-	-	-	−0.33 ^†^	-	-
	2	4.20 ^†^	-	1.47 ^‡^	-	-	-	-	-	-	-
	3	-	-	-	1.33 ^†^	-	−0.61	-	0.38 ^†^	-	-

SDFT: superficial digital flexor tendon; DDFT: deep digital flexor tendon; ALDDFT: accessory ligament of the deep digital flexor tendon; SL: suspensory ligament; LB-SL: lateral branch of the suspensory ligament; MB-SL: medial branch of the suspensory ligament; TA: transverse area; C: circumference; DP: dorsopalmar/plantar length; LM: lateromedial length; ME: mean echogenicity. ^†^
*p* < 0.05; ^‡^
*p* < 0.01; nothing marked *p* < 0.001; only significant differences are demonstrated. * Differences from the subtraction between the mean values of the Mangalarga Marchador and Campeiro breeds (MM–Campeiro).

## Data Availability

All dates are contained within the article.

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
