# Peer review of "Ultrasound Morphometry and Mean Echogenicity of Digital Flexor Tendons, Suspensory Ligament, and Accessory Ligament of Digital Deep Flexor Tendon in Gaited Horses"

_animals, 2023, doi:10.3390/ani13081411_

Round 1

Reviewer 1 Report

Manuscript: animals- 2277561

2.4. Inclusion criteria

was the AAEP scale used for the orthopaedic examination of horses enrolled in the present study? please clarify

2.6. Preparation and ultrasound examination

was the operator imaging experienced delineated? please add details.

were the areas highlighted in line with the current literature? please check the following source: David R. Hodgson, Catherine McGowan and Kenneth McKeever (2013) The Athletic Horse, second Edition. Reproduced with permission of Elsevier.) and please add details.

discussion

the choice to use only one evaluator brings some limitations. Please address them and add to discussion.

image

would be useful to add images related to both horse’s breeds, to help the reader to better visualize morphological characteristics and differences.

Author Response

The authors are grateful for any suggestions made. They have been carefully reviewed and included. All turns made are highlighted in yellow and a point-by-point response is provided below.

-was the AAEP scale used for the orthopaedic examination of horses enrolled in the present study? please clarify. Answer: The AAEP scale of 0-5 was used, this information is included in section 2.5.

-was the operator imaging experienced delineated? please add details. Answer: the information was included.

-were the areas highlighted in line with the current literature? please check the following source: David R. Hodgson, Catherine McGowan and Kenneth McKeever (2013) The Athletic Horse, second Edition. Reproduced with permission of Elsevier.) and please add details. Answer: Yes, we followed the recommendations in the literature for determining the zones. The reference cited by the reviewer is equivalent to those cited in the text. In our article, we standardize the distance between the zones to obtain the best repeatability.

-the choice to use only one evaluator brings some limitations. Please address them and add to discussion. Answer: the limitations were included.

-would be useful to add images related to both horse’s breeds, to help the reader to better visualize morphological characteristics and differences. Answer: the figure was included.

Reviewer 2 Report

Dear authors

Some comments on the attached PDF.

The text does not mention the prevalence of tendonitis/ desmitis pathology in these breeds, which would give relevance to the study, if there are often found during lameness exams?, age related?, activity related? Some insights into this on the breeds presented here would be of relevance.

Also, the ultrasound exams were performed in horses out of training, which limits the data application as it cannot be truly compared to horses under training.

A figure showing the differences in echogenicity would also be worth it.

Author Response

The authors are grateful for any suggestions made, including those in the pdf file. They have been carefully reviewed and included. All turns made are highlighted in yellow and a point-by-point response is provided below.

-The text does not mention the prevalence of tendonitis/ desmitis pathology in these breeds, which would give relevance to the study, if there are often found during lameness exams?, age related?, activity related? Some insights into this on the breeds presented here would be of relevance. Answer: This article is part of a larger project, with a larger number of animals (100), in which the prevalence of lesions was described and analyzed in another article. In this, we seek to identify the ultrasonographic characteristics of sound horses.

-Also, the ultrasound exams were performed in horses out of training, which limits the data application as it cannot be truly compared to horses under training. Answer: We agree with this reviewer and include this in the limitations of the study.

-Line 92: do you mean trotting? Answer: As they are gaited horses, they didn't trotting, so we chose to describe the real gait they did during the exam.

-3.6 section. could this be integrated more in the discussion? as it is more an argumentation of results rather than a presenation of them. Answer: Rereading the section and discussing it with the other authors, we understand that at this point we seek to describe the results, and not a discussion about cause and effect of these results, which is why we chose to keep it.

-A figure showing the differences in echogenicity would also be worth it. Answer: Images demonstrating the levels where the examination was performed have been included and the structures assessed have been included.

Reviewer 3 Report

It was a very nice and interesting article to read; please consider the following edits. Thank you

Line 35: “In Brazil, the Mangalarga Marchador (MM) breed is the most widely breed”. Do you mean widely spread? Please clarify the sentence.

Line 46-47: “Breed variations related to the sonographic characteristics of the digital flexor tendons and palmar metacarpal ligaments are described in horses”. What they found? Tell the readers more about it.

Line 48-49: “However, there are few studies related to tendon and ligament evaluation in gaited horses [10], as well as in the hindlimbs (HL).” What are those studies, what they say and how they are related to your work? Please tell the reader more about them. Expand your introduction, which is too short at the moment. And please cover all the introductory aspects of your study in it.

Line 64-65: “Twenty-five MM horses and 25 Campeiro horses were evaluated and selected to form homogeneous groups regarding body size and weight within the breed and between breeds.” How you calculated the sample size? Can we generalize the results obtained from this sample?

Line 266-267: Please add the ultrasonographic pictures of the studied structure in the manuscript.

Line 439-440: “This is the first study to evaluate and compare the morphometric and ME characteristics of the digital flexor tendons and ligaments of the metacarpal/metatarsal palmar/plantar region between two breeds of gaited horses.” But in the introduction, you said there are studies available on this topic.

Line 476-477: What could be the reasons behind these variations?

Line 497: Why this is so? Please give reasons from the published literature.

Author Response

The authors are grateful for all suggestions made. They have been carefully analyzed and included. All turns made are highlighted in yellow and a point-by-point response is provided below.

-Line 35: “In Brazil, the Mangalarga Marchador (MM) breed is the most widely breed”. Do you mean widely spread? Please clarify the sentence. Answer: The sentence was revised.

-Line 46-47: “Breed variations related to the sonographic characteristics of the digital flexor tendons and palmar metacarpal ligaments are described in horses”. What they found? Tell the readers more about it. Answer: it was included.

-Line 48-49: “However, there are few studies related to tendon and ligament evaluation in gaited horses [10], as well as in the hindlimbs (HL).” What are those studies, what they say and how they are related to your work? Please tell the reader more about them. Expand your introduction, which is too short at the moment. And please cover all the introductory aspects of your study in it. Answer: The section has been expanded.

-Line 64-65: “Twenty-five MM horses and 25 Campeiro horses were evaluated and selected to form homogeneous groups regarding body size and weight within the breed and between breeds.” How you calculated the sample size? Can we generalize the results obtained from this sample? Answer: The final number of animals included in the study. Twenty-five animals per group being a reasonable number to carry out the conclusions that we reach and adapt to other individuals of the breed and species.

-Line 266-267: Please add the ultrasonographic pictures of the studied structure in the manuscript. Answer: Images demonstrating the levels where the examination was performed have been included and the structures assessed have been included.

-Line 439-440: “This is the first study to evaluate and compare the morphometric and ME characteristics of the digital flexor tendons and ligaments of the metacarpal/metatarsal palmar/plantar region between two breeds of gaited horses.” But in the introduction, you said there are studies available on this topic. Answer: This is the first article that describes these characteristics among gaited breeds. The article referred to in the introduction only describes the cross-sectional area of the forelimb tendons of Mangalarga Marcador horses. The sentence has been revised for better understanding.

-Line 476-477: What could be the reasons behind these variations? Answer: The variation we are referring to here is the structure along the limb, which anatomically changes its shape to adapt to function and neighboring structures. These variations were similar between breeds.

-Line 497: Why this is so? Please give reasons from the published literature. Answer: at this point, we only reinforce that the anatomy of these structures is divergent between fore and hindlimbs, at the same level and in the same animal. The text that follows discusses this.

Round 2

Reviewer 1 Report

please revise the English language (minor spelling check)

Reviewer 3 Report

.